# Learning to Live with HIV: The Experience of a Group of Young Chilean Men

**DOI:** 10.3390/ijerph20176700

**Published:** 2023-09-01

**Authors:** Macarena Belén Calderón Silva, Lilian Marcela Ferrer Lagunas, Rosina Cianelli

**Affiliations:** 1Master in Public Health Program, School of Medicine, Pontificia Universidad Católica de Chile, Santiago 833150, Chile; 2School of Nursing, Pontificia Universidad Católica de Chile, Santiago 833150, Chile; 3School of Nursing and Health Studies, University of Miami, Coral Gables, FL 33124, USA; rcianelli@miami.edu

**Keywords:** HIV, young people, qualitative study, hermeneutic, acceptance process, stigma

## Abstract

Young men aged 20–29 present the highest rates of HIV in Chile, yet little is known about their experiences after diagnosis. This study sheds light on the meaning of living with HIV for young gay Chilean males. Qualitative analysis of 11 in-depth interviews, employing Heidegger’s hermeneutical approach, resulted in a depiction of feelings of loneliness and sadness, encounters with discrimination, and a lack of social support. Participants also offer recommendations on utilizing peer support as an essential component for programs targeting young gay men living with HIV in the capital of Chile. Additionally, they suggest that health care providers play a crucial role in supporting the process of accepting the condition. Conclusions: It is imperative to enhance psychological support by integrating it into the HIV program for all clients and families facing crisis situations. The establishment of support groups within hospitals that are part of the national program providers is recommended. Education on comprehensive sexuality should be provided in educational institutions, alongside visible campaigns across all media platforms to dismantle the stigma linked to the disease. Lastly, improvements in care duration and support from health services can be achieved through the provision of comprehensive care founded on unconditional acceptance.

## 1. Introduction

In recent years, the worldwide spread of HIV has been halted and reversed, leading to a reduction in both the incidence and deaths related to Acquired Immune Deficiency Syndrome (AIDS). However, the number of people living with HIV continues to rise, reaching a total of 38 million by 2019 [1]. The global pandemic situation differs from the reality in Chile. The increase in confirmed seropositives between the period of 2010 and 2018 establishes a sustained rise in new HIV diagnoses. The incidence rate in 2018 was 37.5 per 100,000 inhabitants, doubling the rate of 17.5 per 100,000 inhabitants in 2010 [2]. According to the report from the Institute of Public Health (2019), the Metropolitan Region concentrates the highest number of HIV cases between 2010 and 2018, with most new cases occurring in the age group of 20–29 years. During the same period, confirmed HIV cases in men were five times higher than in women [2]. The primary route of transmission is sexual, accounting for 99% of cases, with men who have sex with other men representing 56.7% of reported cases [3]. Since the emergence of the disease, the interpretations attributed to it by individuals and society have been diverse, perpetuating the unfortunate transmission of prejudices in the population. For example, there are notions that it is an immoral disease or the consequence of reprehensible behavior. The sustained social stigma associated with the disease has resulted in the rejection of diagnosed individuals since its description in the 1980s, and this stigma remains prevalent to this day [4].

People living with HIV (PLHIV) are considered a socially vulnerable group due to the consequences of the disease and the subsequent abandonment that often accompanies the diagnosis. The protracted course of the disease, approached as a chronic condition, necessitates a healthcare system that not only grants access to treatment but also provides support and comprehensive professional assistance for an extended period. Presently, this is not being fully realized through the HIV program in Chile, primarily centered on the provision of antiretroviral therapy (ART). Numerous international studies highlight the challenges faced by young individuals when diagnosed with HIV, given the negative emotional states that arise from the diagnosis, such as depression and anxiety. This psychological distress not only leads to a decline in quality of life but also impacts social functioning, correlating with increased involvement in risky sexual behavior and substance use, along with reduced adherence to treatment. Due to the potential adverse psychosocial and medical consequences of an HIV diagnosis in young people, interventions are imperative to help them adapt to this new health condition [5]. In Chile, qualitative studies have been conducted to comprehend the significance of living with HIV. However, one study focused on adults in the AIDS stage, and the other targeted adults of all ages, primarily exploring their personal interpretations of HIV. Consequently, an information gap exists concerning young people. Therefore, a qualitative study was proposed with the objective of revealing the meaning of living with HIV within a group of individuals aged 20–29, as this age group has the highest incidence nationally. The goal is to gather information for impactful interventions that align with the desired UNAIDS (2022) objective of 95–95–95: ensuring that 95% of people living with HIV are aware of their diagnosis, 95% are receiving antiretroviral treatment, and 95% have undetectable viral loads by 2030 [6].

## 2. Method

A qualitative study employing a phenomenological approach, whose analysis was conducted under the hermeneutic perspective of Martin Heidegger, a method that seeks to answer how individuals derive meaning from their lived experiences within a context or when confronted with a specific phenomenon [7]. According to Heidegger, phenomenology allows us to distinguish the stable and permanent from the changing aspects of the world. This is not achieved through meaningless questions, but by attentively listening to expressions of experiences that are not immediately apparent and must be uncovered [8]. Heidegger’s starting point (1962) is the concept of epoche [8]. Strictly speaking, practicing epoche involves setting aside the natural and naturalistic knowledge acquired during the study, to avoid delving into results from similar research conducted at the international level. This is considered in order to preserve the integrity of the ongoing project’s results (theoretical and epistemological adequacy). The inclusion criteria for the study encompassed young men, aged between 20 and 29 years, who had been diagnosed with HIV over a year ago, were undergoing ART treatment, and resided in the Metropolitan Region. Participants were recruited through convenience sampling, primarily utilizing social networks, specifically the “Círculo de Estudiantes Viviendo con VIH (CEVVIH)” (Circle of Students Living with HIV) group. This support group consists of around fifty university students, united in their goal to combat prejudice and effect social and political change regarding HIV on a national level. Following contact with the group administrator, the invitation to partake in the research project was disseminated to the community via “Zoom” meetings. Six young men living with HIV willingly agreed to participate in the study, initiating contact with the responsible researcher through email and returning their signed informed consent through the same medium. This approach had received prior approval from the Ethics Committee of the Pontificia Universidad Católica de Chile. An additional five participants were recruited using the “Snowball” method, through the participation of young men already involved in the study. Data collection employed in-depth interviews as the technique, guided by a single overarching question: “What does living with HIV mean to you?” While the interviews were initially conducted in Spanish, they were later translated into English by an external translator to facilitate publication in an English-language journal. Each interview lasted approximately 45–55 min and was recorded with the interviewees’ consent using two mobile devices. The study did not establish a predetermined number of participants, as the objective was to continue interviewing young people until data saturation was achieved. This occurred after the eighth interview, signifying data credibility. Nonetheless, additional interviews were conducted to verify the results. Given that the research took place amidst the COVID-19 pandemic, some meetings occurred in open spaces such as parks in the eastern sector of Santiago, where a two-meter distance was maintained, and both the interviewer and participant wore masks throughout the interview. Other interviews were conducted online via the Zoom platform. Interviews were coded to ensure participant confidentiality and were transcribed verbatim by the responsible researcher. Each transcription underwent two rounds of review by the same researcher and an additional review by a peer, thereby preserving the authenticity of the interviewees’ discourse and adhering to the qualitative research criterion of “confirmability” [9]. The analysis of the interviews employed Martin Heidegger’s hermeneutic phenomenological analysis technique. Both researchers listened to, read, and analyzed each interview, independently creating their own codebooks. Through Zoom meetings, the process of phenomenological reduction was undertaken, culminating in the identification of the primary units of meaning (consistency). To establish the credibility of the findings, a member check of the main units of meaning was conducted via WhatsApp, involving the participants. Most participants concurred with the findings [10]. Figure 1 depicts the research strategy followed in the current study.

## 3. Results

Among the sociodemographic characteristics of the participant group, the 11 young men were aged between 23 and 29. All of them identified as men who have sex with men. Additionally, all the interviewees had university studies, with 3 of them still pursuing education, while the remaining 8 were employed in various sectors including health, finance, banking, design, dance, and illustration. All 11 participants reside in the Metropolitan Region, with 10 of them working in the public health sector and 1 in the private health sector. Lastly, out of the total 11 young people interviewed, 8 are affiliated with activist groups such as CEVVIH and Acción Gay (Gay Action), while the remaining 3 are not associated with any activist groups.

After performing the phenomenological reduction, the following meaning units were defined because of the in-depth interviews: “What does it mean to live with HIV?”. Table 1, list the different units and its meanings.

“Vision of HIV as something alien”: Since many of them did not have in their collective consciousness the possibility of contracting this sexually transmitted infection.


*YPLHIV: “It didn’t exist in my reality, in my reality there was never going to be HIV, there was never going to be me having any sexually transmitted disease…”*


“Feeling of loneliness”: Mainly during the long process of diagnosis and initiation of ART, distancing themselves from their loved ones for fear of prejudice.


*YPLHIV: “… and at a certain moment I found myself, as if I was alone… with my mother’s prejudices a little bit…not anymore…but at that moment I was alone…”*


“Little acceptance from the family”: A HIV diagnosis is difficult news to process and may make it difficult to assimilate, mainly due to ignorance, prejudice, and stigma surrounding HIV, with young people feeling that their parents have turned their backs on them at a difficult time in their lives.


*YPLHIV: “At some point, they even wanted me to keep away from the family, you know, and that’s how you stop being my son…which is very hard”.*


“Fear of rejection”: Mainly at the time of having to disclose the diagnosis to their respective partners.


*YPLHIV: “…getting to know someone and feeling something more for that person…and having to tell them and being rejected, maybe that has been one of my…fears…”*


“Feelings of anguish, sadness, and guilt”: In the first place related to the diagnosis, where they experience much anguish and despair for not knowing the steps, they must follow to access treatment. Then, in the medium and long term, feelings of grief and sadness arise, but more sporadically. They also feel guilt for not having protected themselves better, for having transmitted the virus to other people or for negative consequences experienced by their nuclear family.


*YPLHIV: “…first of all, I blamed myself a lot for being irresponsible and all that…”*



*YPLHIV: “For me it’s still the issue of…finding a partner or being with someone, for me it’s still stressful…”*


“Need for psychological help”: Given that the whole process that the young people must go through to assimilate the new health condition is complex, and, in most cases, they report having required more psychological support, but for the most part, they did not dare to ask for it.


*YPLHIV: “Having to assume that I was not well, it was hard, and I decided to go about taking that option of the psychologist.”*


“Empowering themselves to educate and support other young people”: All study participants were in the stage of acceptance of their disease, with good adherence to treatment and mostly undetectable. Several young people took the impulse and motivation to guide and support other young people who are starting the same process they have already lived, so that they can better cope with it, and many of them are part of HIV activist groups such as CEVVIH and Acción Gay.


*YPLHIV: “I…I was interested in taking advantage of the space I had to be able to educate other people about sexual rights, because I find that in Chile there is a great lack…”.*


“The importance of peer support”: Especially in the early stages, where many feel alone and have no one to share the diagnosis or clarify doubts. So, they recognize that it is a great help to be part of a group of people who live the same as they do.


*YPLHIV: “I feel like the peer support has been all like…like today I can be sitting here with you and if someone hears me that I have HIV I don’t care…”*


“Self-knowing and loving themselves”: Knowing their bodies, their reactions to medications, they begin to lead a healthier lifestyle and increase their self-esteem.


*YPLHIV: “Living with HIV, as I said, was like a new beginning, a reaffirmation of my self-esteem, my self-love, that I had to take care of myself because it made me feel vulnerable…”*


“Overconcern of the family”: This is because, due to lack of knowledge, some parents continue to associate the diagnosis of HIV with death, feeling great concern about their health checkups or antiretroviral treatment, which the young people themselves consider to be exaggerated.


*YPLHIV: “Well, my mother is still worried…”.*


“Lack of sex education”: That they refer to school education, which had a direct impact on their transmission, and which is maintained after the diagnosis mainly due to the lack of knowledge about protection to prevent transmission to their sexual partners.


*YPLHIV: “I never realized…I realized after I had it, my sexual ignorance about the whole thing…everything about the disease…”*


“Episodes of discrimination by third parties”: Experienced by health care personnel, friends, partners, and even their own family.


*YPLHIV: “…I knew him for about six months or so, and after three months I kind of told him, and it was like hey, I must tell you something, I live with HIV, so he told me…You know what? I can’t be with you…”*


“Lack of health system protection”: Presenting itself on more occasions as a barrier rather than a facilitator for their wellbeing. There is a duality between a poorly functioning system and health personnel who generally provide good care.


*YPLHIV: “Without treatment…I mean what should have taken me one or two months, took me six months, and it was horrible because it was like: Hey! How horrible is this situation, having to go around, having to go around like chasing other people so I could get treatment…”?*


“Lack of continuity in health care”: Which they see reflected in the constant change of the doctor, resulting in great discomfort due to having to tell their complete medical histories to different doctors repeatedly.


*YPLHIV: “…I have already changed at least four times my doc…virologist, and I don’t know, I haven’t felt much continuity either, the only continuous thing I have in my process are my pills…”.*


There are also some emerging codes such as “Deficit in self-care and care of others”, the moment when they stop taking care of themselves or do not do so in an optimal way, either with ART or having risky sexual behaviors, without considering that they can experience recontamination with other strains or transmit the virus to other people.


*YPLHIV: “…but I didn’t care about infecting someone else either…”*


## 4. Discussion

Currently, the HIV promotion and prevention program in Chile has a predominantly biomedical focus, centered on detecting new cases, confirming diagnoses, ensuring adherence to ART, and conducting follow-up examinations. However, a comprehensive approach is lacking [11]. Based on the results obtained from in-depth interviews, to the implementation of new activities within the HIV program in Chile has been suggested. These activities could include providing greater support to young people through motivational interviewing when they start antiretroviral treatment. Motivational interviewing is defined as a collaborative and person-centered approach to guide and strengthen motivation for change, helping individuals adopt healthier behaviors. This approach has also shown to encourage more frequent condom use among young people and to reduce their viral load [12]. Additionally, both scientific evidence and the young people who participated in the study support the idea of including psychological support in the national health program. This support should be available to all users undergoing treatment, as it is not explicitly included in the current HIV program. According to the accounts of the interviewees, only some of them were offered this option. It is important to emphasize that psychological assistance is crucial for individuals living with HIV and their families to cope with the diagnosis. Consultations and sessions with psychologists prove to be fundamental tools in disease treatment. Psychotherapeutic techniques can help users confront the challenges they face due to seropositivity [13]. Based on participants’ feedback, it would be interesting to consider creating support groups in each of the healthcare centers, whether public or private, where HIV treatment is provided. These groups could be led by health professionals such as nurses, midwives, or by other young people who have lived with this condition for more years and have undergone prior training. Research on group therapy has highlighted that a significant aspect of a support group is its ability to demonstrate that individuals are not alone in their diagnosis. They relate to others who share a common experience, fostering a sense of social acceptance and reducing the stigma associated with the disease [14]. In the study by Hays et al., 1990, peers reported that receiving information about HIV, medication, and community services from fellow peers decreased their own fears and uncertainties [15]. The World Health Organization itself, in its document “Providing Peer Support for Adolescents and Young People Living with HIV” (2020), states that peer groups improve adherence, retention, and psychosocial well-being. In all cases, the goal is to provide a source of empathic support and to share positive coping strategies [16]. Families of young people also undergo a complex process upon learning of the diagnosis. Initially, there’s often little acceptance, and some may turn away from their child or relative during a difficult time. This is primarily due to a lack of knowledge about HIV and its implications. However, as time goes by, acceptance usually grows, albeit with a somewhat overprotective stance. At the public health level, it could be valuable to assess the applicability of the Collaborative HIV Prevention and Adolescent Mental Health Program (CHAMP) model. While designed for adolescents living with HIV, it could be extended to young people of the selected ages. The intervention protocol focuses on various aspects including the impact of HIV on families, disease-associated stigma, antiretroviral medication protocols, family communication about sexuality and HIV, parental supervision related to potential risky sexual behaviors, assistance in managing health and medications, and social support and decision-making regarding disclosure [17]. Based on participants’ experiences, it remains evident that living with HIV continues to carry stigma, as many have encountered discrimination at some point. This discrimination can largely be attributed to ignorance and a lack of health education. Reflecting this, the interviewed young people suggest implementing Comprehensive Sexual Education, a concept that has yet to be fully realized in Chile. Sex education is a fundamental right for all, and it should be provided in a comprehensive and impartial manner, covering all authorized alternatives along with their degree of effectiveness. This education empowers individuals to make informed decisions about fertility regulation methods and, importantly, to prevent teenage pregnancy, sexually transmitted infections, and sexual violence and its repercussions. Achieving this requires collaboration between the Ministry of Health and the Ministry of Education, and it should not be left to educational institutions to decide whether to incorporate these topics into the school curriculum, regardless of the values they uphold [18]. Health education must be a cornerstone of the HIV program, with well-defined sessions focusing on self-care regarding ART, responsible sexuality, healthy eating, disclosing the diagnosis, and other relevant topics. Emphasis should be placed on adherence to treatment, given that the likelihood of a person living with HIV remaining on ART after three years is only 57.5% [19]. These educational sessions can be conducted by nurses, midwives, psychologists, or other professionals during the first check-ups with treating infectologists at the respective healthcare centers or hospitals. As indicated by the results concerning the health system, there are criticisms of its responsiveness, particularly its inadequate implementation of effective HIV prevention among young people and adolescents. Additionally, the young people interviewed mentioned the need to shorten the time between the reactive test, confirmation by the ISP (Public Health Institute), and the commencement of ART. Furthermore, instances of discrimination by healthcare personnel are reported. Such practices constitute a serious violation of human rights and raise questions about the integrity of professionals who should adhere to the ethical codes and principles taught during their university training [4]. The healthcare team should be prepared not only for technical and procedural tasks but also to establish genuine and supportive relationships. Coping with the disease often begins with self-help efforts, which are sustainable only up to a point. Subsequently, support from the social network becomes crucial [4]. In conjunction with the proposed interventions for the HIV program, increased participation of people living with HIV (PLHIV) in their health management should be considered. Participation should occur within a context of collaboration between healthcare systems and communities, allowing for greater decision-making power [20]. Finally, it is important to consider assessing health-related quality of life (HRQoL) as part of the follow-up for PLHIV within the program. HRQoL is a multidimensional construct related to an individual’s perception of well-being and level of functioning in various aspects of life, influenced by their health status. Assessment instruments are available to evaluate the impact of HIV and comorbid conditions on HRQoL, such as the EQ-5D and the MOS-HIV [21]. The study’s greatest value lies in gathering insights directly from young people living with HIV, a demographic that had not been thoroughly examined at the national level before. Additionally, conducting the study in Chile, where the HIV situation significantly differs from the rest of the world due to the exponential increase in new cases in recent years, adds to its significance. However, limitations include the homogeneity of the interviewed group in terms of age and affiliation to activist groups. Furthermore, being a qualitative study with a small sample size, the results cannot be generalized.

## 5. Conclusions

Living with HIV for emerging gay youth in the Metropolitan Region brings forth multiple negative implications for both them and their families, which pose individual challenges in coping. Hence, adopting a multidisciplinary approach within the National HIV/AIDS Prevention Program becomes crucial. This approach is necessary considering that medical treatment is already ensured through the prevailing public health policies. This is due to the complex and dynamic journey young people undergo, starting from diagnosis to the eventual acceptance of the disease. This journey demands comprehensive management and holistic support from both the system and healthcare personnel. It is essential to incorporate the measurement of individuals’ quality of life who are part of the program. Nevertheless, there remain areas where users are struggling to find adequate support for their needs. Therefore, conducting new research to develop interventions with the participation of People Living with HIV (PLHIV) would be intriguing. Grasping the concept of participation to foster the empowerment of young people and to facilitate the implementation of the best available strategies in response to user needs could lead to achieving a more comprehensive approach in Chile’s health promotion and HIV prevention program.

## Figures and Tables

**Figure 1 ijerph-20-06700-f001:**
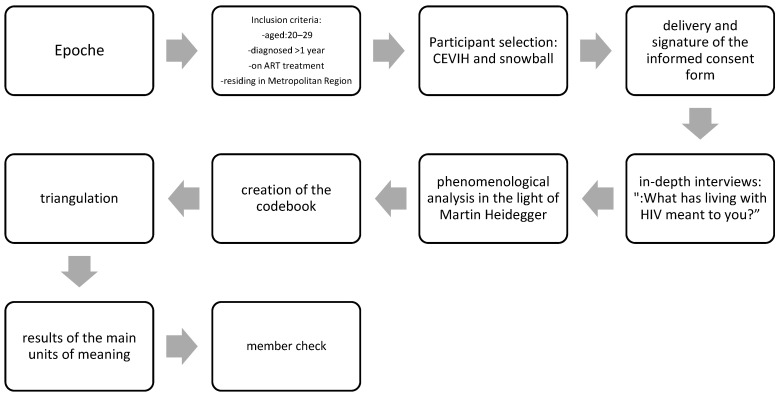
Research strategy diagram.

**Table 1 ijerph-20-06700-t001:** Units of meaning of living with HIV.

Units of Meaning	Literal Quotation	Finding
Vision of HIV as something alien	“You never thought you could touch him”	There is still a lack of knowledge among young people on HIV issues, due to the absence of comprehensive sexual education that should be taught at school.
Lack of Sex Education	“…they never talked about a condom, they never talked about anything…” “…I realized after I had it, my sexual ignorance of all issues, everything that can ward off the disease…”	
Feeling of Loneliness	“…This disease, when you find out [you have it] brings you a…a very great feeling of loneliness…”	HIV diagnosis has negative mental repercussions on both young people and their families; therefore, it is critical to integrate personal and family psychological support into the HIV program.
Feelings of Anguish, Sadness, and Guilt	“First mental chaos that I experienced…I began to feel guilty of my illness, understand?”	
About Family Concerns	“Well, my mom to this day is kind of worried…”	
Little Family Acceptance	“The thing is that three months after I told my mother that I was living with HIV and I was her oldest child and only male child, huh…I think that caused her a lot…a lot of psychological suffering…or serious grief.”	
Empowerment to Educate and Support Other Youth	“…I mean I totally went from super shy, a super introvert and everything, now I feel super powerful…for so many years I saw it was so bad for me, and now I feel that it did nothing bad to me, nothing serious…and it turns your way of seeing life upside down, so I feel empowered, so powerful that I can handle everything…”	Empowerment in a chronic disease is fundamental in the motivation for self-care and consequently viral suppression.
Importance of Peer Support	“I feel that the peer support has been all like…as if today I can be sitting here with you and if someone hears that I have HIV, I don’t care, and I also take pictures and give them visibility…”	The creation of peer groups favors the acceptance of the disease.
Episodes of Discrimination by 3rd Parties	“…It happened to me that…people I was pseudo dating rejected me for having the condition…”	At present, young people continue to be victims of discrimination by partners, friends, and health personnel.
Fear of Rejection	“I don’t know how…to be meeting someone and feel something more for that person…and having to tell them and have them reject me, maybe that has been like my…fear…”	Young people are afraid to disclose their diagnosis to family members and partners for fear of rejection.
Lack of Protection of the Health System	“…how horrible this situation, having to walk as well as chasing other people so they can do the treatment”	Some young people are unhappy with the cumbersome process they must go through to receive antiretroviral therapy.
Lack of Continuity in Health Care	“…They have changed my doctor three times…at that time it was a sensitive issue for me…”	The continuous change of the treating physician that the young people experience is perceived as a difficulty in the continuity of their care.
Self-knowledge and Self-Love	“…so that’s why I’m telling you, it’s really no joke that this issue empowered me and gave me a self-love that I didn’t have before…”	When young people accept their disease and adhere to their treatment, they improve their self-esteem.

## Data Availability

The data is available only by request directly to the corresponding authors.

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
