# Peer review of "Learning to Live with HIV: The Experience of a Group of Young Chilean Men"

_ijerph, 2023, doi:10.3390/ijerph20176700_

Round 1
Reviewer 1 Report
REVIEW REPORT FOR THE STUDY “LEARNING TO LIVE WITH HIV: THE EXPERIENCE OF A GROUP OF YOUNG CHILEAN MEN”
Journal: Int. J. Environ. Res. Public Health
The paper "Learning to live with HIV: the experience of a group of young Chilean men", performs a study with the aim of evaluating carrying out interventions with a greater impact on the health of users, and thus achieve the desired UNAIDS.
Title and summary. The title and abstract express well the object of study, objectives, and results of the article.
Structure of the article. The contents are well organized and they adhere to the IMRaD structure. It includes a theoretical framework of the research.
Focusing on the opportunity of the study, it must be said that it is useful work since it covers one of the major problems resulting from living with the HIV.
Materials and methods.
Regarding the material and methods section, the methodology is tailored to the object of study and the objectives and is explained in a transparent manner while it has been validly applied to guarantee the results.
However, I would like to suggest to the authors, with the intention of reinforcing the choice of methodology, if the authors could use tables and diagrams to present the research strategy and results.
Miles, Matthew B. (2014). Qualitative data analysis : a methods sourcebook. Thousand Oaks, Califorinia :SAGE Publications, Inc.
Results.
The results are significant and they are presented in an adequate and understandable way not only through narration but also with self-explained tables and figures that are also well elaborated in terms of presentation. The results justify and relate to the objectives and methods and the results are of sufficient interest. Also, I suggest to the authors to use diagrams to show the results.
Discussion.
The discussion appropriately compares the study results with other works, highlighting the main study findings.
However, I would propose the inclusion of a bibliographic reference in the discussion section:
Noori, T. et Al., 2022, Health-related quality of life in people living with HIV, publication for the Committee on Environment, Public Health and Food Safety (ENVI), Policy Department for Economic, Scientific and Quality of Life Policies, European Parliament, Luxembourg.
Conclusions
I would suggest that the authors incorporate some further conclusions to their study.
Overall, it is an interesting study and should be considered for publication in Int. J. Environ. Res. Public Health, once the minor revisions proposed have been resolved.
Author Response
Dear,
we are very grateful for your criticism and suggestions, in the first version of the manuscript we had tables attached, so we have decided to reintroduce them in the text, both the research diagram and the table with the results.
We also found the suggested literature on the measurement of quality of life in people living with HIV very interesting, so it is also very well received and we have added it to our article.
Thank you very much.
Best regards
Reviewer 2 Report
We are pleased to have reviewed the article entitled "LEARNING TO LIVE WITH HIV: THE EXPERIENCE OF A GROUP OF YOUNG CHILEAN MEN." This is a complex research study conducted with a population that has been under-studied, and we would like to congratulate the authors on this research.
The introduction is of good quality and effectively presents the research objectives, as well as its position in relation to the existing literature on this topic.
The only suggestion I would like to make to the authors concerns the methodology. Firstly, I encourage them to consider integrating the COREQ-32 (if possible), as this would further establish the credibility of the qualitative method. Secondly, why did the authors choose Martin Heidegger's phenomenological epistemology over that of Husserl, who is considered the father of phenomenology? While it is true that Husserl's approach is more descriptive and transcendental, if we base ourselves on Heidegger's works, it should be descriptive and hermeneutic. The descriptive aspect is absent from the methodology section. Additionally, wouldn't it be preferable to base the method on Colaizzi's (1978) approach?
I have no objection to the method chosen by the authors; however, it is necessary to provide further justification for it.
Regarding the methodology, I believe the interviews were transcribed verbatim, but I couldn't find this information. If it is missing, it should be added. I see that two researchers have read the interviews, indicating that they have indeed been transcribed. This should be included.
The results section is clear and effectively presents the data collected during the interviews.
The discussion is also of high quality and helps to understand the study's objectives and issues.
Author Response
Dear,
we are very greatful four your criticism and suggestions, something that is always necessary for continuous improvement. As a team we decided to use Martin Heidegger's methology mainly because it was a survey of information in a group that has not been studied much in our country. And the Colaizzi 7 step method analyze participants' experiences and identify themes, while the Heidegger method focuses on interpreting texts to get insights and understanding on gaining. we wanted to emphazyse on gaining insights of this scarcely know issue in Chile.
Regarding the COREQ-32 we did not mention in the manuscript, however, when analyzing the step-by-step of out research, we met all the criteria of the checklist.
Finally, in the methodology section, we added that the interviews were transcribed verbatim, as suggested. Again, thank you very much for your comments.
Best Regards
Reviewer 3 Report
The authors present a very well-written and thorough article. However, the discussion could be improved. The discussion lacks an examination of how the current study's findings are similar to or differ from the extant literature. Additionally, there is no examination of the strengths and limitations of the study.
Author Response
Dear,
we are very grateful for your critique on our manuscript, taking your suggestions we added a little more literature in our discussion and extended a little more our conclusion to deliver a more complete work.
Thank you very much.
Best regards